# Barriers, enablers and acceptability of home-based care following elective total knee or hip replacement at a private hospital: A qualitative study of patient and caregiver perspectives

**Jason A. Wallis**[1,2]*, **Emma Gearon**[1,2], **Justine Naylor**[3,4], **Kirby Young**[5], **Shay Zayontz**[6], **Phillipa Risbey**[5], **Ian A. Harris**[4,7], **Rachelle Buchbinder**[1,2], **Denise O'Connor**[1,2]

1 Department of Epidemiology and Preventive Medicine, School of Public Health and Preventive Medicine, Monash University, Melbourne, Victoria, Australia, 2 Monash-Cabrini Department of Musculoskeletal Health and Clinical Epidemiology, Cabrini Health, Malvern, Victoria, Australia, 3 Whitlam Orthopaedic Research Centre, Liverpool Hospital, Sydney, New South Wales, Australia, 4 South Western Sydney Clinical School, University of New South Wales, Sydney, New South Wales, Australia, 5 Centre for Allied Health Research and Education, Cabrini Health, Melbourne, Victoria, Australia, 6 Department of Orthopaedic Surgery, Cabrini Health, Melbourne, Melbourne, Victoria, Australia, 7 Institute for Musculoskeletal Health, The University of Sydney, Sydney, New South Wales, Australia

* jwallis@cabrini.com.au

**Data Availability Statement:** Interview data from this study are not available for sharing given potential confidentiality implications associated

## Abstract

### Background

To facilitate implementation of home-based care following an elective total knee or hip replacement in a private hospital, we explored patient and caregiver barriers and enablers and components of care that may increase its acceptability.

### Method

Thirty-one patients (mean age 71 years, 77% female) and 14 caregivers (mean age 69 years, 57% female) were interviewed. All themes were developed using thematic analysis, then categorised as barriers or enablers to uptake of home-based care or acceptable components of care. Barrier and enabler themes were mapped to the Theoretical Domains Framework.

### Results

Eight themes emerged as barriers or enablers: feeling unsafe versus confident; caregivers' willingness to provide support and patients' unwillingness to seek help; less support and opportunity to rest; positive feelings about home over the hospital; certainty about anticipated recovery; trusting specialist advice over family and friends; length of hospital stay; paying for health insurance. Five themes emerged as acceptable components: home visits prior to discharge; specific information about recovery at home; one-to-one physiotherapy and occupational therapy perceived as first-line care; medical, nursing and a 24/7 direct-line

with smaller participant samples and consistent with current ethical approvals. The Institutional Review Board is the Cabrini Research Governance Office (CRGO). The website is https://www.cabrini.com.au/research-and-education/research-ethics/. The non-author point of contact is researchgovernance@cabrini.com.au.

**Funding:** The author(s) received no specific funding for this work.

**Competing interests:** The authors have declared that no competing interests exist.

perceived as second-line care for complications; no one-size-fits-all model for domestic support. Theoretical domains relating to barriers included emotion (e.g., feeling unsafe), environmental context and resources (e.g., perceived lack of physiotherapy) and beliefs about consequences (e.g., unwillingness to burden their caregiver). Theoretical domains relating to enablers included beliefs about capabilities (e.g., feeling strong), skills (e.g., practising stairs), procedural knowledge (e.g., receiving advice about early mobility) and social influences (e.g., caregivers' willingness to provide support).

## Conclusions

Multiple factors, such as feeling unsafe and caregivers' willingness to provide support, may influence implementation of home-based care from the perspectives of privately insured patients and caregivers. Our findings provide insights to inform design of suitable home-based care following joint replacement in a private setting.

## Introduction

Following acute care elective inpatient admission for total knee and hip replacements (TKRs, THRs), patients may receive either ongoing inpatient care at a rehabilitation facility/ward or non-inpatient care. Inpatient rehabilitation includes 24-hour a day nursing and medical care, daily sessions of physical and occupational therapy, and may be indicated for a minority of patients with complex conditions, the elderly, and those without social supports [1]. Utilisation of this rehabilitation pathway is common amongst elective TKR or THR recipients in Australia [2], but for the majority, non-inpatient care is suitable [3–6]. Hospital-at-home and rehabilitation-at-home facilitate earlier transition to home where patients may continue to receive one-to-one acute care (i.e., hospital-at-home) and/or rehabilitation-at-home [7, 8]. Compared to inpatient care, hospital-at-home and rehabilitation-at-home following TKRs and THRs provides similar patient outcomes including no increased risk of hospital readmission [3–7]. Hospital-at-home and rehabilitation-at-home may also provide superior patient satisfaction without increasing caregiver burden and may lead to a reduction in costs to the health service [7, 9].

In Australia, over 111,000 TKRs and THRs occur annually within two distinct healthcare sectors [10]. Public hospitals are a single-payer system with patient costs covered by the government and private hospitals form part of a multiple-payer system for insured or self-funded patients. The majority (70%) of joint replacements are performed in the private sector in Australia [11]. Despite evidence of similar outcomes, a higher proportion of patients in the private sector in Australia utilise more costly inpatient rehabilitation compared to the public sector (56% and 33% in the private sector versus 7% and 4% in the public sector for TKRs and THRs respectively) [2]. Shifting care from the inpatient setting to the home setting may raise concerns for both patients and their caregivers, especially in private settings where rates of inpatient rehabilitation are higher. It is also likely that perverse incentives exist in private settings in Australia that encourage higher uptake of inpatient rehabilitation [2]. This highlights the need to target the private sector where this problem is largest. Given rates of joint replacement are expected to substantially increase during the next decade [11], demand for inpatient rehabilitation is likely to increase, especially in the private sector in Australia.

Qualitative research can provide an understanding of factors influencing implementation and acceptability of home-based care services by giving insights into people's perceptions that

cannot be achieved by other research methods [12]. Previous qualitative studies have provided some understanding of these factors within the context of different countries [13–16] but these may not be applicable to private hospital settings in Australia. One previous qualitative study exploring clinician and patient decision-making for rehabilitation following THR and TKR in private hospital settings in Australia identified patient preference for inpatient rehabilitation as a key barrier to home discharge [17]. However, this study did not use a theoretical approach to explore barriers and enablers to home-based care in the private setting. Using a theoretical framework facilitates a more thorough assessment of the factors influencing implementation and can assist in designing tailored strategies likely to achieve greater uptake of hospital-at-home and rehabilitation-at-home [18–20].

From the perspective of privately insured patients undergoing an elective TKR or THR and their caregivers in a private hospital setting, this qualitative study aimed to (i) explore barriers and enablers to implementation of home-based care, including theoretical explanations, and (ii) explore components of home-based care that may increase its acceptability for privately insured patients. In this study we used the term implementation to describe the uptake of best care, evidence-based practices into routine practice with the aim of improving patient care [21]. We used the term acceptability to reflect the extent to which people receiving a healthcare intervention consider it to be appropriate, based on anticipated or experienced cognitive and emotional responses to the intervention [22].

## Materials and methods

### Design

A qualitative descriptive study involving semi-structured interview questions aimed at exploring patients' and caregivers' perspectives on home-based care was conducted [23, 24]. Ethics approval was obtained from the Cabrini Health Human Research Ethics Committee prior to commencement of the study (02-28-10-19). All patients and caregivers provided written, informed consent prior to their participation in the interview. The consolidated criteria for reporting qualitative research (COREQ) checklist was used for study reporting [25].

### Participants

Privately insured patients with osteoarthritis who were scheduled for an elective TKR or THR between November 2019 and March 2020 at a single private institution were eligible to participate in the study. Adult caregivers who would primarily help with activities of daily living (e.g., meals preparation) during the individual's recovery were also eligible to participate in the study. Patients and caregivers not able to speak fluent English were excluded.

### Setting

This study was conducted at a large 508-bed, not-for-profit, private hospital (Cabrini Health, Melbourne, Australia) which conducts a high volume of elective orthopaedic surgery. Following surgery, acute inpatient care was typically provided for up to five days and then patients were referred to either rehabilitation-at-home, outpatient rehabilitation or inpatient rehabilitation. Within the multiple payer system, patients may be eligible for some or all services depending on their insurance coverage. At the time of study recruitment, hospital-at-home was not utilised for this patient cohort and provides an alternative option to acute inpatient care. For hospital-at-home, patients remain admitted as acute patients where they have a short acute inpatient stay (2 days) followed by daily care from a physician, nurse, and physiotherapist (all employed by the hospital) in their homes for a limited time (typically 2 to 3 days).

These patients are then referred onto rehabilitation-at-home or outpatient physiotherapy following discharge from hospital-at-home.

For rehabilitation-at-home, patients are discharged from acute care to the hospital service ('Therapy in the Home' with staff employed by the hospital) or an external private home rehabilitation service. Rehabilitation-at-home programs are provided for about 4 to 6 weeks by allied health professionals and include negotiated goals aiming to optimise the patient's functioning and quality of life, caregiver support and education. For outpatient rehabilitation, patients may attend a facility such as a private physiotherapy practice (may include home visits) or an outpatient rehabilitation service. For inpatient rehabilitation, patients receive subacute care in the dedicated rehabilitation ward or an external hospital, and length of stay is typically about 10 days. Following inpatient rehabilitation, patients are commonly referred to outpatient rehabilitation at either the hospital outpatient facility or to an external facility.

## Recruitment

We invited participation in the study at consecutive preadmission information sessions. These group-based sessions were conducted weekly at the hospital and face-to-face. A hospital-employed clinician presenter informed attending patients and caregivers about the study verbally, and provided a study flyer in patients' information packs. If any patients and caregivers were interested in participating, they returned an expression of interest or contacted the primary investigator (JW) available at the end of each session or via phone. Interested patients and caregivers were provided with the participant information and consent forms by the primary investigator. The primary investigator explained the purpose of the research was 'to gain an understanding of the potential obstacles, enablers and acceptability with alternative rehabilitation services to inpatient rehabilitation, such as home-based care'.

At this preadmission session, patients completed the Risk Assessment and Prediction Tool (RAPT) [26]. The RAPT is a 6-item questionnaire with a 12-point scoring system used to predict a patient's destination at discharge following acute care, with an additional open-ended question asking the patients preferred destination. The RAPT is also used to identify the need for targeted interventions (e.g., educating patients and families) to increase patient readiness and self-confidence for discharge [27]. The predictive accuracy of the score has been validated in different countries and settings with predictive accuracy around 75% in an Australian public hospital setting and 78% in a US setting [27]. Incorrect RAPT predictions mainly occur for scores between 6 to 9, the intermediate risk category that predicts discharge home with additional intervention such as rehabilitation-at-home [27]. Lower scores ($< 6$) are indicative of higher risk for inpatient rehabilitation [27]. To ensure a range of views were explored in this study, purposive sampling was used to include patients with their preferred discharge destination as either inpatient or home using the RAPT. We planned to recruit 30 patients undergoing TKR or THR and 15 caregivers on the basis that this would likely enable us to reach data saturation with no new themes emerging.

## Data collection

The semi-structured interview guide was informed by the Theoretical Domains Framework (TDF) [19, 20]. The order of questions in the interview guide was flexible and adapted by the interviewer (e.g., 'tell me more about that') to fit the flow of conversation with patients and caregivers. We explored the theoretical domains considered most relevant to the uptake of home-based care [18] (S1 and S2 Tables). The interviews also explored components of care that may improve acceptability of home-based care to patients and caregivers. An

understanding of which components of care (e.g., medical care), including domestic supports (e.g., transport, cleaning, meal preparation), are most important to patients and their caregivers, may also inform interventions to increase uptake of hospital-at-home and rehabilitation-at-home services by privately insured patients. The interview guide was developed by our multidisciplinary team with clinical expertise in joint replacement surgery and rehabilitation, as well as academic expertise in qualitative and implementation research to ensure it was clinically relevant and comprehensible. The same interview questions, including prompts, were used for all interviews, and field notes were taken during and after interviews. The interview guide was not pilot tested prior to data collection.

Interviews were conducted face-to-face in patient's homes or via the telephone, and conducted one-to-one or in dyads (i.e., patient with their caregiver) depending on their preferences. The interviews were conducted before or after surgery to include patients and caregivers with and without recent experience of different postoperative pathways (e.g., inpatient rehabilitation and rehabilitation-at-home). All interviews were conducted by a male postdoctoral research fellow (JW), an experienced physiotherapist (20 years) providing care at the hospital and with in-depth knowledge of the hospital services and experience in leading qualitative research, including conducting one-to-one interviews. None of the patients or caregivers were known to the interviewer. All interviews were audiotaped, transcribed verbatim by an independent transcriber, and read by the interviewer to check for accuracy of transcription. Individual transcripts were sent back to each patient and caregiver to confirm veracity, and to make any necessary changes.

At interview the following patient data were collected: age, sex, body mass index, employment status, comorbid conditions, RAPT score, and previous joint replacement surgery. For patients who had already undergone joint replacement, we also collected inpatient length-of-stay (acute and rehabilitation), insurance provider, surgeon, discharge destination and any post-operative adverse events. For caregivers, we collected their age, sex, relationship to the patient and employment status.

## Data analysis

NVivo 12.0 data management software was used to assist with thematic analysis [28]. Qualitative analysis of interview data commenced with a close review of each transcript by two authors, both of whom are experienced in qualitative data collection and analysis (JW, EG). Working independently, each author developed descriptive codes using an inductive approach to code the interview data. Descriptive codes were identified from five initial interviews (3 patients, 2 caregivers) followed by a discussion between the two authors to reach consensus. This method was repeated for a further five and then 10 randomly selected interviews. The remaining 19 interviews were coded by one researcher (JW).

All nine authors discussed the emergent themes until a consensus on themes was reached. Subsequently, themes were categorised deductively as either barriers or enablers to implementation of home-based care, and acceptable components of home-based care. We used the concept of 'barriers and enablers to implementation' as the themes related to patients' uptake of home-based care. The barrier and enabler themes were then mapped to the TDF by one author (JW) and checked by a second author (DOC). For acceptability, these themes related to how patients and caregivers considered the components of home-based care to be appropriate, based on anticipated or experienced cognitive and emotional responses to the intervention. There was no theoretical underpinning for the themes related to acceptability. All authors have knowledge that hospital-at-home and rehabilitation-at-home interventions are underutilised, yet with similar effectiveness compared to inpatient care.

## Results

All interviews were undertaken between November 2019 and March 2020. Data saturation was achieved after 39 interviews (20 face-to-face, 19 via telephone) involving 31 patients and 14 caregivers. The patient response rate to participate was 26%. All patients who expressed interest participated in the interviews, except for one female patient who was ineligible due to having a partial knee replacement. Twenty interviews were conducted before surgery (13 patients, five caregivers and two pairs of patients and caregivers interviewed concurrently). Nineteen interviews were conducted one to four weeks after surgery (12 patients, 3 caregivers and four pairs of patients and caregivers interviewed concurrently). The average interview length was 27 minutes (range 15 to 45 minutes). No repeat interviews were conducted.

### Participants

Table 1 and S3 Table reports the characteristics of the 31 patients. Mean (SD) age was 71 (10) years, 24 patients were female, and 24 patients had caregiver assistance. Twenty patients had a planned TKR, and 11 patients had a planned THR. One patient interviewed prior to a planned TKR cancelled surgery. Twenty-nine patients had a RAPT score of six or above. Of those that had the surgery, 13 patients received inpatient rehabilitation and the remaining 17 patients received rehabilitation-at-home, and none received hospital-at-home. Eleven of the 13 patients who received inpatient rehabilitation had a RAPT score between 6 and 9, while all 17 patients who received rehabilitation-at-home had RAPT scores of 6 or above. Six out of 18 patients who had indicated a preference for inpatient rehabilitation before surgery received rehabilitation-at-home, and all had a RAPT score of 10 or above.

For the 14 caregivers, mean (SD) age was 69 (12) years, 8 were female, 5 were employed and 10 were spouses or partners. Other caregivers included a sibling, daughter, friend, and paid caregiver. One caregiver interview was excluded from the qualitative analysis due to non-fluent English.

### Summary of barrier and enabler themes

Eight themes that emerged as barriers and enablers of home-based care included: feeling unsafe versus confident; caregivers' willingness to provide support and patients' unwillingness to seek help; less support and opportunity to rest; positive feelings about home over the hospital; certainty about anticipated recovery; trusting specialist advice over family and friends; length of hospital stay; paying for health insurance. A summary of the eight themes is included below, with illustrative quotes. S4 Table shows the themes mapped to theoretical domains, with additional illustrative quotes.

**Feeling unsafe versus confident.** Patients who received inpatient rehabilitation expressed multiple safety concerns if they had opted for home-based care. This included concerns about mobility and personal care (e.g., fear of falling in the shower), managing household activities, managing stairs, wound care, taking medications or injections, causing damage to the new joint, and not having a live-in caregiver if something was to happen when they were alone.

> "I live alone, and I've got stairs, and until I regain strength and mobility, I would be better off in rehab, than on my own."

> (Patient 2, female, 51 years, RAPT score 8, interviewed before surgery, received inpatient rehabilitation)

**Table 1. Patient characteristics.**

| Variables | All patients (n = 31*) | Received inpatient rehabilitation (n = 13) | Received rehabilitation at home (n = 17)$ |
|---|---|---|---|
| | Mean (SD) | Mean (SD) | Mean (SD) |
| Age, years | 71 (10) | 74 (11) | 68 (8) |
| Acute inpatient length of stay, days | 4.5 (1.8) | 4.9 (2.0) | 4.3 (1.5) |
| Inpatient rehabilitation length of stay, days | 12.4 (3.1) | 12.4 (3.1) | N/A |
| | N (%#) | N (%^) | N (%^) |
| Interviewed before surgery* | 15 (48) | 8 (53) | 6 (40) |
| Female | 24 (77) | 11 (46) | 13 (54) |
| Employed | 12 (39) | 4 (25) | 8 (75) |
| Surgery type | | | |
| TKR* | 20 (65) | 9 (45) | 10 (55) |
| THR | 11 (35) | 4 (36) | 7 (64) |
| Previous joint replacement | | | |
| Knee | 8 (26) | 4 (50) | 4 (50) |
| Hip | 2 (6) | 1 (50) | 1 (50) |
| Caregiver | | | |
| Living with caregiver | 21 (68) | 5 (24) | 16 (76) |
| Caregiver assistance available* | 3 (10) | 1 (33) | 1 (33) |
| No caregiver available | 7 (22) | 7 (100) | 0 (0) |
| RAPT score | | | |
| 1–5 | 2 (6) | 2 (100) | 0 (0) |
| 6–9 | 18 (58) | 11 (61) | 7 (39) |
| 10–12* | 11 (35) | 0 (0) | 10 (91) |
| Preferred discharge destination | | | |
| Inpatient rehabilitation | 18 (58) | 12 (67) | 6 (33) |
| Home* | 13 (42) | 1 (8) | 11 (85) |
| Orthopaedic surgeon | | | |
| Surgeon 1 | 11 (35) | 6 (55) | 5 (45) |
| Surgeon 2* | 10 (32) | 3 (38) | 6 (75) |
| Surgeon 3 | 8 (26) | 3 (38) | 5 (63) |
| Surgeon 4 | 2 (6) | 1 (50) | 1 (50) |

*Surgery cancelled (n = 1);

#Proportion of patients for each variable per total number of patients;

^Proportion of patients receiving either inpatient rehabilitation or rehabilitation-at-home per number of patients for each variable.

$Seven patients received Cabrini's service, and 10 patients received an external service including Medibank-at-home (n = 3), Remedy (n = 3), unknown home service provider (n = 4); RAPT—Risk Assessment and Prediction Tool—score 1 to 5 predicts transfer to inpatient rehabilitation, score 6 to 9 predicts additional intervention to discharge directly home, score 10 to 12 predicts discharge directly home; TKR = total knee replacement; THR = total hip replacement

Patients who received rehabilitation-at-home felt safe and relieved any anxiety at home by having an able caregiver who was retired or living with them, especially if the caregiver had a healthcare background and could assist in their management. Patients also felt safe if they perceived themselves (or caregiver perceived the patient) as young, fit, and not needing full-time care. Other factors that increased confidence at home included rearranging their house, having ready-made food available for their household and pets, having built up strength pre-surgery, being motivated to exercise or receiving advice that exercise was simple to perform, receiving reassurance that they could contact their surgeon if there were surgical complications, and having a trusted physiotherapist who would assist in their care. For a few patients and

caregivers interviewed after surgery, having received training on crutches and stairs under supervision of the physiotherapist, and having received information about the effectiveness and safety of early walking changed their beliefs and concerns about safety with mobility.

> "My husband actually is a medical practitioner so that may have relieved any anxiety I might have had if I've gone home without having that support. I had that confidence nothing is going to go really wrong here."

> (Patient 27, female, 65 years, RAPT score 12, interviewed before surgery, received rehabilitation-at-home)

**Caregivers' willingness to provide support and patients' unwillingness to seek help.** Caregivers expressed a willingness to being 'on-call' when they were needed, and without limits to the amount of support they would provide.

> "Whatever she needed, I would look after her but, I didn't have a limit or an expectation"

> (Caregiver 7, female, daughter, 40 years, interviewed after surgery, patient received rehabilitation-at-home)

Patients who received inpatient rehabilitation did not feel comfortable asking family for support, expressed concerns about their caregiver being unreliable, giving up their own activities and hobbies, or keeping them awake at night. A few female patients interviewed before surgery also thought it would be too taxing or unrealistic for their husbands to be performing household tasks from making beds, gardening, and cooking.

> "You're putting an enormous amount of pressure on the people around you who have to support you. I don't want to punish the people around me."

> (Patient 26, male, 63 years, RAPT score 8, interviewed after surgery, received inpatient rehabilitation)

**Less support and opportunity to rest.** Patients who were interviewed either before or after surgery, and received inpatient rehabilitation described this setting as more 'official', 'disciplined' and 'sensible' representing potential barriers to home-based care. This included the perception of more intensive supervision and exercise sessions, better facilities and equipment, 'on-tap' medical care for careful monitoring, analgesia and managing complications. There were concerns about feasibility of receiving the same intensity of rehabilitation support with home-based care, and concerns about potential delays or absence of clinical support at home.

> "They probably wouldn't supervise you that much because they'd only be here for half an hour."

> (Patient 16, female, 73 years, RAPT score 7, interviewed after surgery, received inpatient rehabilitation)

Some patients preferred inpatient rehabilitation to 'receive a break', 'switch off' from their usual daily lives and activities, and 'relax' knowing their meal would be provided, and they did not have to cook and clean, which would not be the case with home-based care.

"I need a rest—I'd like to be not cooking and I just want to switch off."

(Patient 22, female, 68 years, RAPT score 9, interviewed before surgery, received inpatient rehabilitation)

**Positive feelings about home over the hospital.**   Patients and caregivers perceived their homes in a positive way and the hospital in a negative way (even if patients preferred inpatient rehabilitation). Homes were perceived as familiar, relaxing, comfortable, allowing better sleep, more visitors, independence, control, solitude, and greater flexibility to develop their own routines.

"I'd much rather be at home in my own bed."

(Patient 15, female, 68 years, RAPT score 10, interviewed before surgery, received rehabilitation-at-home)

Hospital wards were perceived as noisy and busy places, akin to an 'institution' or a 'prison' that provided terrible food that did not meet their dietary requirements or cultural preferences. Hospitals had routines that made them go 'stir crazy', feel uptight, bored, and depressed, and provided group sessions that were the same for everyone.

"I kept thinking in the first week, 'What would I have done if I'd been there [inpatient rehabilitation]? I would have been in prison."

(Patient 31, female, 74 years, RAPT score 8, interviewed after surgery, received rehabilitation-at-home)

For a few patients interviewed after surgery, a negative hospital experience changed their preference in favour of rehabilitation-at-home, and a few patients interviewed at the beginning of the Covid-19 pandemic perceived the hospital was a place where there was a risk of catching the contagious virus.

"The only time I might change my mind about [rehabilitation-at-home] is if you told me corona virus, Ebola virus, was rampant through hospitals."

(Patient 26, male, 63 years, RAPT score 8, interviewed after surgery, received inpatient rehabilitation)

**Certainty about anticipated recovery.**   Patients who preferred home-based care, perceived the home setting facilitated the 'optimal recovery', provided one-on-one rehabilitation attention, and psychological benefits resulting from being in their own home and from outdoor physical activity. Patients' determination for an 'optimal recovery' included reaching milestones in the shortest possible time by performing their normal routines, activities, hobbies, and work.

"I thought by having physio come into my home they would arrive at a set time, rather than being in a group and not having that one-on-one attention"

(Patient 11, female, 65 years, RAPT score 7, interviewed after surgery, received rehabilitation-at-home)

"I really want to be on my feet and doing what I should do in as short a time as possible."

(Patient 10, female, 73 years, RAPT score 9, interviewed after surgery, received rehabilitation-at-home)

Patients and caregivers who had previously had joint replacement surgery preferred the same rehabilitation setting (and sometimes the same clinician). This gave them confidence in the knowledge that this service provided a safe and effective recovery. Some patients and caregivers who had not previously had joint replacement surgery were uncertain with post-operative mobility and in a 'totally new world' by not knowing if they would be well enough to cope at home. Instead, they preferred to wait and see how they recovered after surgery before making a choice about rehabilitation setting.

"The outcome's unknown at the moment for me. I don't know what he will do, how my leg might be. Everything might go haywire."

(Patient 25, female, 73 years, RAPT score 7, interviewed before surgery, received inpatient rehabilitation)

**Trusting specialist advice over family and friends.**   Receiving a recommendation from a specialist or general practitioner about their discharge destination was perceived as being helpful as doctors were experts and knew about their health circumstances. For some patients this allowed them to be open-minded and their doctor's advice influenced their preferred rehabilitation setting. Receiving advice from friends and family about their rehabilitation destination, including receiving "convincing lectures" was not always trusted by patients.

"I'm still open to the fact that if I'm having difficulty post-surgery and they suggest that I go to rehab ["Oh, he'll go" (Caregiver 11, female, wife, 55 years)] "Then I'll go"

(Patient 20, male, 60 years, RAPT score 12, interviewed together before surgery, received rehabilitation-at home)

"Whilst people are very free with advice and well-meaning, that's what happened historically and is useless information today."

(Patient 3, male, 70 years, RAPT score 11, interviewed before surgery, received rehabilitation-at-home)

**Length of hospital stay.**   For some patients, a short acute inpatient length of stay (i.e., 3 days) was perceived to be too short, acting as a potential barrier to hospital-at-home. A longer acute hospital stay (i.e., 5 days) or the flexibility to extend their stay was perceived to enable sufficient time to recover from surgery and acted as a potential enabler for rehabilitation-at-home. In contrast, some patients preferred the shortest possible hospital stay, acting as potential enabler for hospital-at-home. The usual length of stay in inpatient rehabilitation (i.e., 10 days) also represented a potential enabler for rehabilitation-at-home.

"When I learnt that I would have had to stay there [inpatient rehabilitation] seven days or ten and I didn't want to do that. I would have been happy to go for three [days], but not for seven or ten."

(Patient 31, female, 74 years, RAPT score 8, interviewed after surgery, received rehabilitation-at-home)

**Paying for health insurance.**   A few patients and caregivers expected their health insurer to cover their preferred setting, were 'not impressed' that home services depended on their level of insurance. A few patients also felt their insurance payments warranted inpatient rehabilitation, representing a potential barrier to home-based care.

"He has private insurance and all covered. Why to send him home?"

(Caregiver 3, female, paid caregiver, 55 years, interviewed before surgery, patient received inpatient rehabilitation)

A few patients and caregivers expressed a willingness to pay for rehabilitation-at-home and expressed satisfaction with hospital staff who advocated on their behalf for the insurance company to fund services at home serving as potential enablers for home-based care.

"We're in a position where we're fortunate where we can pay [for home-based care] now."

(Caregiver 5, female, sibling, 73 years, interviewed before surgery, patient received rehabilitation-at-home)

## Summary of acceptability themes

Five themes that emerged as acceptable components of home-based care included: home visits prior to discharge; specific information about recovery at home; one-to-one physiotherapy and occupational therapy perceived as first-line care; medical, nursing and a 24/7 direct-line perceived as second-line care for complications; no one size fits all model for domestic support. A summary of the five themes and illustrative quotes is included below.

**Home visits prior to discharge.**   Some patients and caregivers expressed a desire for a home visit by a physiotherapist or occupational therapist prior to discharge for reassurance their home was a suitable and safe environment for rehabilitation. This included advice on safe access around their home (especially kitchens and bathrooms), to assess their need for equipment (e.g., suitable sized shower stools), and to identify suitable places to perform their exercise routines (e.g., walking programs).

"I was surprised somebody didn't come to check out the house prior to coming home to make sure it's suitable."

(Patient 17, female, 64 years, RAPT score 21, interviewed after surgery, received rehabilitation-at-home)

**Specific information about recovery at home.**   Patients and caregivers expressed the importance of specific information about recovery at home (e.g., expected recovery of pain and activity levels each week; extent to which the caregiver would need to look after the patient; rehabilitation-at-home schedules). Some patients and caregivers also wanted more information about 'strong' analgesics for managing pain overnight when the pain was perceived to be worse, and more practical instructions for self-administering injections to prevent deep vein thrombosis.

"You need some more information about the issues that you might come across at home."

(Patient 24, female, 72 years, RAPT score 8, interviewed after surgery, received rehabilitation-at-home)

"So, people can know what's normal and put people's minds at ease"

(Caregiver 7, female, 40 years, daughter, interviewed after surgery, patient received rehabilitation-at-home)

**One-to-one physiotherapy and occupational therapy perceived as first-line care.**
Patients and caregivers desired at least two home visits from clinicians in the first week following home discharge, with a higher number of sessions desired if patients were living alone and without caregiver support. For subsequent weeks, it was acceptable to reduce or tailor the number of regular visits depending on the patient's progress. Important components of care included provision of information and advice in response to patient and caregiver questions, observing how patients interacted in their home setting, and ensuring exercise performance was 'absolutely right', 'customised', and diverse with a range of exercises that were progressed incrementally. One-to-one sessions were perceived to provide support to manage individual difficulties (e.g., pain levels), individual challenges (e.g., exercise performance), and to improve confidence to walk at home, as well as improve their mood, and alleviate safety fears about returning home.

"You would want somebody [physio] maybe two or three times a week in the first week and then dropped it down if they're coping well."

(Patient 15, female, 68 years, RAPT score 10, interviewed before surgery, received rehabilitation-at-home)"

"Seeing how the patient was interacting with their home environment, with the animals, with showering, kitchen, all of those sorts of things which they wouldn't necessarily see [in hospital]"

(Caregiver 9, female, friend, 75 years, interviewed after surgery, patient received inpatient rehabilitation)

**Medical, nursing and a 24/7 direct-line perceived as second-line care in the case of complications.** Patients believed routine home visits by doctors and/or nurses were not essential, providing there was an option for a doctor and/or nurse visit if required. This included a 24-hour, 7 days-a-week direct line for medical advice and to arrange a home visit if required. This option of a doctor and/or nurse visit was perceived as important in the immediate rehabilitation period given the patients and caregivers' perception that it would be difficult and costly to attend a private emergency department. This was also perceived to alleviate concerns about pain management, wound care, blood clot prevention and managing potential medical complications such as adverse reactions to medications. Some patients and caregivers believed it was acceptable if medical support was provided by the patient's general practitioner, a doctor that was part of the home service or a nurse practitioner.

"You don't really need a doctor unless you are in trouble."

(Patient 2, female, 51 years, RAPT score 8, interviewed before surgery, received inpatient rehabilitation)

"At three o'clock in the morning, [patient] was screaming in pain and I give her Palexia or something and about an hour later, she's still screaming. What do I do? A direct line to the doctor to tell me what to do"

(Caregiver 13, male, spouse, 75 years, interviewed after surgery, patient received inpatient rehabilitation)

**No one size fits all model for domestic support.**   Acceptable components of domestic support included a balanced healthy main meal (with multicultural options), and a cleaner for 'hard' chores such as vacuuming, laundry, cleaning the bathroom and changing the bedding. For patients without a caregiver and tentative about getting in the shower and falling, personal care support was desired to ensure they were safe and stable to get dried and dressed. There was not a 'one size fits all model' due to different needs and depending on recovery. An option to be able to apply for, or modify, the domestic support was also important, providing there were no out-of-pocket costs.

"You can't just make one size fits them all you know"

(Patient 9, female, 68 years, RAPT score 9, interviewed after surgery, received rehabilitation-at-home)

"The heavy cleaning, the vacuuming, washing the floors, toilets because you can't do it"

(Patient 24, female, 72 years, RAPT score 8, interviewed after surgery, received inpatient rehabilitation)

For patients with a preference for inpatient rehabilitation and without a caregiver, domestic support was perceived as 'unrealistic' because it did not replace a caregiver who would perform regular tasks throughout the day and did not address their specific barriers to care at home. For some patients with a preference for rehabilitation-at-home, domestic support was perceived not to be required because of existing services (e.g., cleaners, online food, and shopping services), having food prepared in the freezer, and their caregiver's proficiency to assist the patient at home (e.g., cooking and driving).

"Not for me—I have got a cleaner who comes every two weeks and [Husband] does everything"

(Patient 12, female, 67 years, RAPT score 10, interviewed after surgery, received rehabilitation-at-home)

## Discussion

Fears around safety (e.g., being home alone), perceived lack of rehabilitation support and opportunity to rest at home, and patients' unwillingness to seek help from caregivers emerged as major barriers to uptake of home-based care as perceived by privately insured THR and TKR patients. These barriers were commonly perceived by patients who received inpatient rehabilitation. Theoretical domains relating to these barriers included 'emotion',

'environmental context and resources' and 'beliefs about consequences'. A live-in caregiver, positive perceptions of their home (e.g., freedoms) and fitness, caregiver willingness to support the patient's choice, and advice from specialists emerged as major enablers to uptake of home-based care. Theoretical domains relating to these enablers included 'beliefs about capabilities', 'skills', 'procedural knowledge' and 'social influences.'

The key barrier and enabler themes identified in our study converged with themes identified in qualitative studies conducted in multiple countries, hospital settings (public and private) and with different aims [13–17]. For example, a perceived lack of caregiver support and lack of confidence to cope with daily activities emerged as a barrier for home discharge in a study that examined patients' choice of discharge destination in Singapore [16]. 'Paying for health insurance' that emerged as a barrier to home-based care in our study is consistent with findings from a previous Australian study where patients' perceived sense of entitlement influenced the discharge destination towards inpatient rehabilitation [17]. However, most patients and caregivers in our study did not think financial factors influenced their preferred rehabilitation setting. It is possible that they were more focussed on their safety and the 'optimal recovery' than financial considerations. It is also possible patients and caregivers were not willing to divulge to the interviewer that financial factors influenced their preferred care.

Another unique finding in our study was that the patients' preferred choice of care setting may be altered by external threats. This included the perceived increased risk with inpatient care as a direct result of the Covid-19 pandemic. A recent study conducted in the same hospital setting observed reductions in inpatient rehabilitation following THR and TKR that appeared to indicate a change in patient preference as a direct result of the pandemic [29]. Another unique finding was caregivers' willingness to support the patient's choice for rehabilitation-at-home or inpatient rehabilitation, and to provide support to patients during rehabilitation-at-home. This contrasted with patients' unwillingness to seek help from caregivers in our study. A previous qualitative study that examined the experiences of live-in caregivers (spouses and offspring) in looking after patients during hospital-at-home following a joint replacement has shown that caregivers may provide a small degree of assistance with hygiene, dressing, mobility and overall responsibility [15]. A review of hospital-of-home has also shown a low level of burden experienced by caregivers after joint replacement [7]. Our findings suggest that clinicians should encourage their patients to ask family or friends to be their caregiver in the early days at home and this may be an important enabler to home-based care following THR or TKR. As well as providing assurance to the patient about the overall low level of burden [7] and small degree of assistance their caregivers may need to provide [15].

Patients and caregivers in our study had a high desire for home physiotherapy including information about recovery at home. Domestic support was desired for patients without caregiver support, and medical support was deemed needed in the case of complications. Hospital-at-home and rehabilitation-at-home services worldwide primarily involve nursing and allied health services, with care sometimes involving physicians and home help for various conditions including TKRs and THRs [7]. The increased desire for home physiotherapy in the immediate rehabilitation period identified in our study may have been because patients were concerned with the 'optimum recovery' of their joints and returning to their normal activities as quickly and safely as possible. The need for home physiotherapy and to obtain knowledge about recovery at home was also found in previous qualitative studies exploring the experiences with rehabilitation-at-home after THR in China [13]. Domestic support may have been a lower priority in our setting because most patients interviewed in our study had caregivers and they were an affluent population. In view of these findings, hospital-at-home and rehabilitation-at-home services need to consider an optimal balance between medical, allied health and domestic support.

Our findings from a single centre private hospital give valuable insight for health systems and providers designing suitable home programs that may be applicable to other Australian private hospitals with similar patient and clinical characteristics. We have also provided a thick description of the patients and caregivers, study context, and research process so that readers can judge if our findings from a private institution are transferable to their setting. The role of the policy maker is to ensure financing, guidelines and objectives for home care services reduce demand for inpatient hospital beds, reduce costs and optimise health outcomes. The role of the clinician is to provide patients and caregivers with the knowledge, skills, resources, and confidence they require to engage in early discharge, hospital-at-home and rehabilitation-at-home. An understanding of our study findings may help to illuminate factors (i.e., barriers and enablers) influencing uptake of home-based care and provide health systems and clinicians with a framework for successful implementation.

Our findings also give valuable insight into designing implementation strategies likely to improve uptake of rehabilitation-at-home or hospital-at-home in the private hospital setting. A 2018 Cochrane review containing 18 randomised controlled trials showed patient education and information interventions (e.g., patient information materials) probably improves recommended healthcare delivery by 11% and 12% respectively, compared to usual care [30]. However, none of these studies were targeting uptake of home-based models of care. Patient education and information interventions could target important theoretical domains (e.g., emotion), and address identified barriers (e.g., dispel concerns about home safety), and enhance the enablers (e.g., trusted specialists communicating the evidence for recovery) of home-based care. These interventions could be presented using various modes of delivery (e.g., print materials, videos, pre-operative education with knowledgeable health professionals). They could be targeted to specific patients who prefer inpatient rehabilitation, as indicated by a lower RAPT score (i.e., 6 to 9), those living without a caregiver and retired. Targeting healthcare staff to address patients' choice of care and sense of entitlement may also be warranted based on qualitative studies conducted in private settings in Australia [17]. Other barriers related to having adequate caregiver support, or home-based care not being perceived as equivalent support to inpatient care may be more difficult to address. Therefore, not all patients may be willing to receive rehabilitation-at-home and hospital-at-home programs in private hospital settings.

Strengths of this study included the purposive sampling of patients and caregivers with preferences for care at home and in hospital ensuring we captured these differing perspectives. We also explored theoretical explanations for low uptake of home-based care that can be used to design implementation interventions aimed at improving uptake in private hospital settings. The response rate for patient participation may be underestimated as we did not know if all patients attending the preadmission sessions were eligible, and the caregiver response rate was not calculated. The interview guide did not explicitly differentiate between hospital-at home and rehabilitation-at-home, the perspective from patients having received hospital-at-home and their caregivers are missing and the interview guide was not pilot tested. As we did not include questions for every TDF domain in the interview guide, only those considered most relevant, we cannot exclude the possibility that factors associated with these domains do not exist. Also, our study sample was limited to patients and their caregivers from a private setting, and we did not explore manager, clinician, or policy level perspectives. Views of administrators across multiple private hospitals and whether they see home-based care as beneficial or a lost opportunity to increase revenue is an area of further research. A risk of social desirability bias exists as patients may have not wanted to divulge to the interviewer all factors relevant for the preferred care, and caregivers could have given the impression that they were supportive of the patient's choice.

## Conclusions

Multiple factors emerged, such as feeling unsafe and caregivers' willingness to provide support, that influence implementation of home-based care from the perspectives of privately insured patients and caregivers. Our findings give valuable insight for health systems and providers designing suitable home programs. Future research is needed to investigate strategies that dispel fears around safety and promote the benefits of home-based care for improving uptake of these services for patients undergoing a TKR or THR in a private hospital setting.

## Supporting information

**S1 Table. Patient interview schedule.**
(DOCX)

**S2 Table. Caregiver interview schedule.**
(DOCX)

**S3 Table. Additional patient characteristics.** *Surgery cancelled (n = 1); #Proportion of patients for each variable per total number of patients; ^Proportion of patients receiving either inpatient rehabilitation or rehabilitation-at-home per number of patients for each variable.
(DOCX)

**S4 Table. Barriers and enablers of home-based care, with themes mapped to the Theoretical Domains Framework and illustrative quotes.** P—patient; C—caregiver; F—female; M—male; B—interviewed before surgery; A—interviewed after surgery.
(DOCX)

**S1 Checklist. Consolidated criteria for reporting qualitative research (COREQ) checklist.**
(PDF)

## Acknowledgments

The authors thank all the patients and caregivers giving their time to participate in this study.

## Author Contributions

**Conceptualization:** Jason A. Wallis, Emma Gearon, Justine Naylor, Kirby Young, Shay Zayontz, Phillipa Risbey, Ian A. Harris, Rachelle Buchbinder, Denise O'Connor.

**Data curation:** Jason A. Wallis.

**Formal analysis:** Jason A. Wallis, Emma Gearon, Denise O'Connor.

**Investigation:** Jason A. Wallis.

**Methodology:** Jason A. Wallis, Emma Gearon, Justine Naylor, Kirby Young, Ian A. Harris, Rachelle Buchbinder, Denise O'Connor.

**Project administration:** Jason A. Wallis.

**Writing – original draft:** Jason A. Wallis.

**Writing – review & editing:** Jason A. Wallis, Emma Gearon, Justine Naylor, Kirby Young, Shay Zayontz, Phillipa Risbey, Ian A. Harris, Rachelle Buchbinder, Denise O'Connor.

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
