## [Decision Letter · Decision Letter 0]

17 Jan 2022

PONE-D-21-32322Barriers, enablers and acceptability of hospital-at-home and rehabilitation-at-home following elective total knee or hip replacement at a private hospital: A qualitative study of patient and caregiver perspectivesPLOS ONE

Dear Dr. Wallis,

Thank you for submitting your manuscript to PLOS ONE. After careful consideration, we feel that it has merit but does not fully meet PLOS ONE’s publication criteria as it currently stands. Therefore, we invite you to submit a revised version of the manuscript that addresses the points raised during the review process.

We look forward to receiving your revised manuscript.

Kind regards,

Carsten Bogh Juhl, PhD

Academic Editor

PLOS ONE

Journal Requirements:

Additional Editor Comments (if provided):

Dear Author

The reviewer have found your study interesting - but have some important suggestions - especially clarifying the method and especially align the conclusion with the available data as these only represent a selected group of patients

Best Academic editor Carsten Juhl

Reviewers' comments:

Reviewer's Responses to Questions

**Comments to the Author**

1. Is the manuscript technically sound, and do the data support the conclusions?

Reviewer #1: Yes

Reviewer #2: Partly

Reviewer #3: Partly

Reviewer #4: Yes

2. Has the statistical analysis been performed appropriately and rigorously? 

Reviewer #1: N/A

Reviewer #2: Yes

Reviewer #3: N/A

Reviewer #4: N/A

3. Have the authors made all data underlying the findings in their manuscript fully available?

Reviewer #1: No

Reviewer #2: No

Reviewer #3: Yes

Reviewer #4: No

4. Is the manuscript presented in an intelligible fashion and written in standard English?

Reviewer #1: Yes

Reviewer #2: Yes

Reviewer #3: No

Reviewer #4: Yes

5. Review Comments to the Author

Reviewer #1: This paper identified barriers and enablers of hospital-at-home and rehabilitation-at-home in privately insured patients receiving total knee and hip replacement. Furthermore, components of care that may increase the acceptability of hospital-at-home and rehabilitation-at-home were identified.

Overall, this study provides an interesting insight into barriers, enablers and acceptability of hospital-at-home and rehabilitation-at-home. The authors used interviews for exploring privately insured patients’ and caregivers’ perspectives and used triangulation in both the data collection and in the analysis. The authors presented the findings in a thick description and related the findings to findings in other studies in the discussion.

However, some concerns way be raised related primarily to the introduction and the method section. These concerns must be addressed to increase the credibility of this study.

• The authors mentioned that hospital-at-home and rehabilitation-at-home facilitates earlier transition to home and provides similar results compared to inpatient care. Furthermore, the demand for inpatient rehabilitation in the private sector will increase, but it is mentioned in the discussion, that it is unknown whether the administrators of private hospitals sees hospital-at-home and rehabilitation-at-home as beneficial. Therefore, the reason for carrying out this qualitative study in a private setting can be mentioned more explicitly.

• The authors used the Consolidated criteria for reporting qualitative studies (COREQ) checklist for their reporting (line 101-102) which is a relevant checklist for qualitative studies. Several points related to the research team and reflexivity are not fully described in the paper.

• The authors used interviews for exploring patients’ and caregivers’ perspectives. On this basis, I assume, that a phenomenological approach was taken, however, the approach is not described in the paper. Please identify the approach adopted to this qualitative study.

• The authors obtained study approval and the informants gave signed informed content prior to commencement of the study. It is recommended to add this information to the manuscript.

• In the selection process of informants purposive sampling was used (line 123-124) – a relevant approach for achieving an insight into the phenomenon of interest. The objective is to explore barriers and enablers of both rehabilitation-at-home and hospital-at-home (line 92-95), but only the selection criteria: inpatient rehabilitation and rehabilitation-at-home were used (line 123-124). As reported in the results none of the informants received hospital-at-home, thus the perspective from patients receiving hospital-at-home and their caregivers are missing.

• The interview guide was guided by the Theoretical Domains Framework (line 129-131). Using a framework facilitates a thorough assessment of a phenomenon of interest in this case barriers and enablers of hospital-at-home and rehabilitation-at-home. Not all domains of the Theoretical Domains Framework were covered by questions in the interview guide (Table S1, S2) and a reason for this is not described in paper. Furthermore, it is recommended and usual practice to pilot test the interview guide. It is left unclear whether the interview guide has been pilot tested in this study (line 129-138). It would be useful to know have more information about this.

• The patients were invited to participate in the study at the preadmission information session (line 122-123), but in what way they were invited and how many patients refused to participate is unclear.

• The interviews were undertaken between November 2019 and March 2020 (line 182), but please describe the setting in which the interviews took place.

• The discussion summarizes the finding and mention influential theoretical domains for the barriers (discussion, line 5-7). It is also suggested to mention the influential theoretical domains after summarizing the enablers (discussion, line 7-10).

• In the fourth section of the discussion the authors wrote: “Patients and caregivers in our study had a low desire for medical and domestic support at home and a greater desire for home physiotherapy….”. The last four quotes in Table 3 indicates a desire for domestic support at home under the theme “No one size fits all model for domestic support” to increase the acceptance of hospital-at-home and rehabilitation-at-home indicates a bit of a contrast. It is suggested to rephrase.

• The authors aimed at exploring the perspectives of privately insured patients at a private hospital and only privately insured patients participated in this study. It is recommended to mention in the conclusion (page 29) that these are privately insured patients.

• Some minor suggestions for the tables:

- Table 1 – Caption: please chose either patient or participant.

- Table 1 – Surgery type: 20 patients having total knee replacement were interviewed. Nine patients received inpatient rehabilitation and 10 patients received rehabilitation-at-home (= 19 patients). I presume, that the * should be placed at TKR and not at “Interviewed before surgery”.

- Table 1 – Please provide an explanation for the abbreviations TKR and THR.

- Table 2 – I have a bit of a hard time seeing which quotes belong to which theoretical domain in a few sections for example in the section “Feeling unsafe” - can it be made more manageable?

- Table 2 – The first section includes quotes from patient receiving inpatient rehabilitation. But in what way can the patient who cancelled surgery (last quote) be an inpatient? Suggest doing it as the authors did in the section: “Less support and opportunities to rest” (last quote).

Reviewer #2: Thanks for the opportunity to read your article and make a review. This article addresses a topical issue in terms of patients and caregivers’ perspectives on rehabilitation-at-home and provides information about barriers and enablers for this intervention. It is important to consider participants’ views on the intervention, as acceptability may undermine the uptake and thus the implementation. Strengths of this study includes the qualitative approach that is underutilized, the inclusion of both patients and caregivers as well as use of TDF.

The manuscript is well written and the presentation comprehensive and detailed, but there are some points that should be clarified. The authors should clarify definitions and differences between the different interventions and concepts around which the article circulates. This will help to provide an understanding of the authors’ preconceptions as well as clarify the difference between a statement classified as an enabler and a statement classified as a theme about an acceptable component.

In the introduction inpatient rehabilitation is described, but rehabilitation-at-home and hospital-at-home is not elaborated to the same extent although the components are central to the study. Knowing the components of the interventions and the extent to which patients and caregivers’ have knowledge of the interventions and their content, is important to assess the interviewees statements and for assessing the relevance to their own practice for the international reader. The components of the interventions may be described briefly in the section about setting and be detailed in an additional file.

The authors should elaborate and/or discuss hospital-at-home, as it is unclear whether hospital-at-home is studied, since the intervention is sparsely described, the interview guides (S1 and 2) use the term “home-based care”, it is unclear if the patients have knowledge about the intervention and no patients receive hospital-at-home (which is reflected in tables 1 and S3 and furthermore the purposive sampling doesn’t include hospital-at-home participants (p.6, l 122-124)). The authors may consider adding and discuss information on whether hospital-at-home and rehabilitation-at-home are interventions that are offered to all patients undergoing surgery in the private / public sector in Australia (or the private hospital where the study takes place).

The second aim of the study concerns patients’ experience of components influencing the acceptability of the rehabilitation-at-home and hospital-at-home. The authors may consider to accurately describe acceptability and components influencing acceptability – what is the rationale for selection of the specific components (e.g., information and support by health professionals that are components asked about in the interview guide) and is it theoretical underpinned (data collection p. 7, l 133-138)?

Implementation is used inconsistently in literature and across disciplines, thus an elaboration in this study is requested especially as implementation is a part of the conclusion. It is uncertain whether implementation is seen primarily as uptake to the interventions, as this is the primary argument for the purpose of the study. Does it e.g., primarily focus on the mechanism through which delivery is achieved.

The authors should clarify why the use of data from RAPT is important in this qualitative study – do the authors expect it to have an impact on patients’ and/or caregivers’ statements or the interpretation of these? The authors might include a discussion if the RAPT results were as expected and whether it has had an impact on the statements / interpretations (P. 7, l. 147- and p. 9, l. 193-).

Title and conclusion: The title and conclusion are appropriate if the above comments on acceptability and hospital-at-home can be substantiated in the article.

Minor details to consider:

P. 6, l. 107-: Consider using the term caregivers to clarify the transition from patient to caregiver

P. 8, l.155-166: Missing source at last line.

P. 8, l. 158: Body mass index is part of table1 and not mentioned in results or discussion – consider relevance or moving to S3.

P. 8, l. 175: “All authors” – is it a reference to all 9 authors?

P. 14, table 2, first quote under the heading “Intentions”: The patient uses the term “the only time I might change my mind about RAH is if you told me corona virus…” – are those the words that support the result that “the hospital was a place where there was a risk of catching the contagious virus”?

P. 25: In the first lines of the discussion “TKA or TKA” probably should be corrected to “THA and TKA”

Reviewer #3: A language revision is recommended throughout the manuscript. Please check grammar and spelling - there are several typos in the manuscript. E.g. a typo in the second line in the discussion section, “TKR or TKR” one should be THR.

Reviewer #4: Comments to the authors:

Overall, the manuscript is well-written with detailed information about methodological approach and considerations. Please see the comment below regarding each manuscript section.

Abstract:

• The abstract clearly states the rationale for conducting the study, the aim, results and conclusion in a precise way that inclines me to further reading.

Introduction:

• The induction gives a clear description of the patient population, health care setting and potential benefits of at-home rehabilitation.

• The aim of the study is clear, however, it would be informative to include the authors hypothesis, as this will strengthen the transferability of the study.

Methods:

• The sampling strategy seems appropriate according to the study aim.

• Data collection; there is a need for additional information regarding data collection, including the interview setting, i.e. face-to-face, hospital setting, at home, telephone? Was the setting the same for all participants, or was is based on the participants’ preferences?

• It was nice to read that validations of transcripts were made by the participants.

• The interviewer’s role related to the participants was not stated. Was the interviewer known to the participants before the interview? This is crucial due to risk of bias, pre-understanding, and influence during the research process. Should also be included in the discussion section.

• Line 129: It is stated that the interviews are semi-structured. The supplementary material show that the interview guide is quite comprehensive and includes very specific questions. Suggest describing the interview procedure more in detail related to use of the interview guide, and/or referring to the interviews as “structured”?

Results:

• To increase transferability of the study, please provide a description of the recruitment process (i.e. description of number of patients invited, declined - and reasons for declining).

• The sample size seems to be realistic and sufficient to answer the research question. The sample consisted of a variability regarding sex, employment status, surgery type and caregiver status.

• The result section is very comprehensive. The use of both Tables (Table 2 + 3) and text description leads to many repetitions. It is recommended to merge the sections, and to use citations in the texts, as this will lead to a more coherent results section.

• Line: 192 + 193. Suggest reporting only the numbers or the percentages.

• Table 2: Under theme “Patient unwillingness to seek help”. The citation “ I have a husband at home but he’s a busy person, I mean he will in around but the idea of him doing all that I do is unrealistic” does not really seem to support the theme about being unwilling to seek help.

Discussion:

• The discussion includes a comprehensive summary of the findings, which is unnecessarily long. Instead more critical discussion of the findings related to existing evidence is needed.

• Please provide discussion related to the transferability of the results, including application of results from a single-center private institution.

6. PLOS authors have the option to publish the peer review history of their article (what does this mean?). If published, this will include your full peer review and any attached files.

Reviewer #1: No

Reviewer #2: **Yes: **Tenna Askjaer

Reviewer #3: No

Reviewer #4: No

---

## [Author Response · Author response to Decision Letter 0]

28 Apr 2022

Due to the length of the response and formatting, please refer to the attached document 'Response to reviewers'.

---

## [Decision Letter · Decision Letter 1]

9 Jun 2022

PONE-D-21-32322R1Barriers, enablers and acceptability of home-based care following elective total knee or hip replacement at a private hospital: A qualitative study of patient and caregiver perspectivesPLOS ONE

Dear Dr. Jason Wallis

Thank you for submitting your manuscript to PLOS ONE. After careful consideration, we feel that it has merit but does not fully meet PLOS ONE’s publication criteria as it currently stands. Therefore, we invite you to submit a revised version of the manuscript that addresses the points raised during the review process.

We look forward to receiving your revised manuscript.

Kind regards,

Carsten Bogh Juhl, PhD

Academic Editor

PLOS ONE

Journal Requirements:

Reviewers' comments:

Reviewer's Responses to Questions

**Comments to the Author**

1. If the authors have adequately addressed your comments raised in a previous round of review and you feel that this manuscript is now acceptable for publication, you may indicate that here to bypass the “Comments to the Author” section, enter your conflict of interest statement in the “Confidential to Editor” section, and submit your "Accept" recommendation.

Reviewer #1: All comments have been addressed

Reviewer #2: (No Response)

Reviewer #3: All comments have been addressed

Reviewer #4: All comments have been addressed

2. Is the manuscript technically sound, and do the data support the conclusions?

Reviewer #1: Yes

Reviewer #2: Yes

Reviewer #3: Yes

Reviewer #4: Yes

3. Has the statistical analysis been performed appropriately and rigorously? 

Reviewer #1: N/A

Reviewer #2: N/A

Reviewer #3: N/A

Reviewer #4: N/A

4. Have the authors made all data underlying the findings in their manuscript fully available?

Reviewer #1: (No Response)

Reviewer #2: No

Reviewer #3: No

Reviewer #4: Yes

5. Is the manuscript presented in an intelligible fashion and written in standard English?

Reviewer #1: Yes

Reviewer #2: Yes

Reviewer #3: Yes

Reviewer #4: Yes

6. Review Comments to the Author

Reviewer #1: Thank you for the opportunity to re-review your article.

You have adequately addressed the comments I have raised in the previous round of review leading to an improved manuscript that I think is suitable for publication.

Reviewer #2: Thank you for the revised document. The previous raised comments have been taken into account.

P. 24, l. 525-526 and l. 536-537: Patients' data (patient, gender, time for interview and intervention) are listed differently than in the rest of the article.

Reviewer #3: Thank you for letting me review this revised manuscript. The manuscript (especially the method section) is much more accurate and transparent in the description after the revision. However, I still have some minor issues to point out.

Abstract

According to the PLOS One author guideline the length of the abstract should not exceed 300 words – your abstract is 389 word. Please, revise this.

Methods

Researchers characteristics

Please elaborate how much experience the interviewer has in qualitative research.

Results

In the result section I would prefer that the author summed up/listed the barriers and enablers themes found, instead of listing them in a supplementary table (S4). Afterall, the barrier and enabler themes are the result of the study. The same goes for the acceptability themes – list the themes in the result section instead of in a supplementary table (S5).

Listing the themes in the result section, may prevent the difference in themes listed in the abstract from those listed as headlines in the result section (they are not typed exactly in the same way) and they are not listed in the same order.

This issue is also seen in the acceptability themes.

After each quotation it would have been nice also to know the participant´s age and RAPT score, to get a quick impression of the participant´s mobility

In the last two quotations under the headline “paying for health insurance” what kind of rehabilitation did these patients receive?

Under the headline “no size fits all model for domestic support” what does P9 stand for? And Please do also elaborate (P12, F, A, RAH).

Discussion

In the discussion section (line 571-579), where you discus the contrast between patients´ unwillingness to seek help from caregivers and caregivers willingness to support the patients. In this path you refer to a study (ref 15) is this a comparison with your study? – if so please elaborate the finding of this study (ref 15) and compare them to your findings and e.g. state if it is an enabler or barrier to home-based care/rehabilitation.

Conclusions

In your conclusion, could you please be more specific about what is “multiple factors”?

What do you want your readers to remember from reading this paper? Any specific factors?

Please remember to align this with the conclusion in the abstract.

Reviewer #4: Thank you for the opportunity to review the article again. The authors have answered all my questions and comments, and have satisfactorily made appropriate changes to the manuscript.

I have only a few suggestions for correcting potential errors:

Please check the wording in the following line. Page 8, lines 185: "we invited participation in the study at consecutive group-based, face-to-face preadmission information sessions, conducted weekly at the hospital".

Page 10, lines 237-239: participants x 3 (written as "patients" in all other places).

7. PLOS authors have the option to publish the peer review history of their article (what does this mean?). If published, this will include your full peer review and any attached files.

Reviewer #1: No

Reviewer #2: No

Reviewer #3: No

Reviewer #4: No

---

## [Author Response · Author response to Decision Letter 1]

18 Jul 2022

Please refer to the attached file 'Response to reviewers'

---

## [Decision Letter · Decision Letter 2]

9 Aug 2022

Barriers, enablers and acceptability of home-based care following elective total knee or hip replacement at a private hospital: A qualitative study of patient and caregiver perspectives

PONE-D-21-32322R2

Dear Dr Wallis

We’re pleased to inform you that your manuscript has been judged scientifically suitable for publication and will be formally accepted for publication once it meets all outstanding technical requirements.

Kind regards,

Carsten Bogh Juhl, PhD

Academic Editor

PLOS ONE

Additional Editor Comments (optional):

Reviewers' comments:

Reviewer's Responses to Questions

**Comments to the Author**

1. If the authors have adequately addressed your comments raised in a previous round of review and you feel that this manuscript is now acceptable for publication, you may indicate that here to bypass the “Comments to the Author” section, enter your conflict of interest statement in the “Confidential to Editor” section, and submit your "Accept" recommendation.

Reviewer #3: All comments have been addressed

2. Is the manuscript technically sound, and do the data support the conclusions?

Reviewer #3: Yes

3. Has the statistical analysis been performed appropriately and rigorously? 

Reviewer #3: N/A

4. Have the authors made all data underlying the findings in their manuscript fully available?

Reviewer #3: Yes

5. Is the manuscript presented in an intelligible fashion and written in standard English?

Reviewer #3: Yes

6. Review Comments to the Author

Reviewer #3: Thank you for the opportunity to re-review the article. The authors have answered my questions and comments and have satisfactorily made appropriate changes to the manuscript.

I have only two minor points to raise.

Table 1: please, align the use of n=xx (not N=xx)

To generalize the study results, please consider the male representation in the study.

The authors have interviewed 31 participants of which 24 is female ~ 77 %. The study includes 27 quotations of which 23 is made by female participants ~ 85%. I suggest that the authors add a few more quotations coming from male participants.

7. PLOS authors have the option to publish the peer review history of their article (what does this mean?). If published, this will include your full peer review and any attached files.

Reviewer #3: No

---

## [Editor Report · Acceptance letter]

15 Aug 2022

PONE-D-21-32322R2 

Barriers, enablers and acceptability of home-based care following elective total knee or hip replacement at a private hospital: A qualitative study of patient and caregiver perspectives 

Dear Dr. Wallis:

I'm pleased to inform you that your manuscript has been deemed suitable for publication in PLOS ONE. Congratulations! Your manuscript is now with our production department. 

Kind regards, 

on behalf of

Dr. Carsten Bogh Juhl 

Academic Editor

PLOS ONE